# Revisiting Multi-Omics Data to Unravel Galectins as Prognostic Factors in Head and Neck Squamous Cell Carcinoma

**DOI:** 10.3390/biomedicines12030529

**Published:** 2024-02-27

**Authors:** Oriana Barros, Vito Giuseppe D’Agostino, Lucio Santos, Rita Ferreira, Rui Vitorino

**Affiliations:** 1Department of Medical Sciences, Institute of Biomedicine iBiMED, University of Aveiro, 3810-193 Aveiro, Portugal; orianamirandabarros@gmail.com; 2Experimental Pathology and Therapeutics Group, Research Center of IPO Porto (CI-IPOP)/RISE@CI-IPOP (Health Research Network), Surgical Department of Portuguese Oncology Institute of Porto (IPO Porto)/Porto Comprehensive Cancer Center (Porto. CCC), 4200-072 Porto, Portugal; llarasantos@gmail.com; 3Department of Cellular, Computational and Integrative Biology (CIBIO), University of Trento, Via Sommarive 9, 38123 Trento, Italy; vito.dagostino@unitn.it; 4LAQV-REQUIMTE, Department of Chemistry, University of Aveiro, 3810-193 Aveiro, Portugal; ritaferreira@ua.pt; 5UnIC, Department of Surgery and Physiology, Faculty of Medicine, University of Porto, 4200-319 Porto, Portugal

**Keywords:** galectins, prognosis, multi-omics, head and neck squamous cell carcinoma

## Abstract

Head and Neck Squamous Cell Carcinoma (HNSCC) is a malignant cancer with a poor prognosis. Galectins (Gal) have been the subject of intensive research, but the comparative prognostic value of each Gal type is not yet understood. Therefore, a literature search for evaluating galectins as prognostic biomarkers in HNSCC was conducted. The relationship between Gal expression in HNSCC with HPV and *TP53* mutational status was assessed using the UALCAN database. The impact of these biomarkers on prognosis was analyzed using ToPP and CPPA web tools. The expression of galectins in the tumor microenvironment and the impact on prognosis depending on the cancer immune subtype were analyzed using single-cell RNA sequencing. *Gal-1* and *Gal-3BP* were shown to be promising biomarkers with a triple function for the prediction of HPV and *TP53* mutational status, stratification of the HNSCC prognosis, and prediction of the response to treatment. In addition, these two galectins have been shown to be most influenced by the tumor microenvironment of HNSCC. *Gal-1* and *Gal-3BP* are the most promising galectins in HNSCC. Furthermore, this study highlights the need for further studies to evaluate galectins in HNSCC and clarify the role of individual Gals in the patient’s stratification.

## 1. Introduction

Head and neck squamous cell carcinoma (HNSCC) is one of the most common malignant cancers worldwide and originates from the epithelial cells of the oral cavity, pharynx, and larynx. Currently, the prognosis is based on Tumour Node Metastasis (TNM) Cancer Staging. The p16 is the only approved prognostic biomarker applied in oropharyngeal cancer. In other subtypes of HNSCC, p16 is not mandatory [1,2]. Still, there is a lack of diagnosis tools to optimize risk stratification in HNSCC.

Galectins are a family of non-integrin β-galactoside binding lectins encoded by *LGALS* genes. According to their structure and the number of carbohydrate-recognition domains, they can be classified into prototype, chimera type, and tandem repeat type. Galectins are involved in numerous biological processes including the tumor-stroma crosstalk, which is considered a critical hallmark that promotes the progression and metastasis of solid tumors. These lectins are key players in modulating the tumor microenvironment (TME) and in regulating tumor progression, invasion, and metastatization processes. The TME consists of several cell types, namely, T cells, B cells, natural killer cells, macrophages, neutrophils, dendritic cells, stromal cells, endothelial cells, cancer-associated fibroblasts, adipocytes, and stellate cells. These cells interact with non-cellular components of the TME, such as extracellular matrix components and exosomes that also participate in carcinogenesis, invasion, and tumor metastasis [3,4]. The TME often induces immunosuppressive states that affect the efficacy of conventional chemotherapy treatments. Thus, in recent years, therapies targeting TME constituents have been developed to be administered alone or in combination with conventional chemotherapy treatments to increase their efficacy. To identify patient subsets that may benefit from immunotherapy, it is necessary to identify the predictive biomarkers of treatment response to optimize the therapeutic management of patients with HNSCC [3,4].

Several works have been done to evaluate the role of specific galectins in the prognosis of HNSCC [5,6,7,8,9,10,11,12,13,14,15,16,17,18,19,20,21,22,23,24,25,26,27,28,29,30,31,32,33,34,35,36,37,38,39,40,41,42]. However, there are no studies that make a systematized evaluation of galectins in HNSCC to identify the panel of galectins with greater interest to be validated as a prognostic biomarker. This work aimed to identify the best panel of galectins that can be used as biomarkers of HPV and *TP53* mutational status, prognosis, and response to treatment in patients with HNSCC. To evaluate which galectins could have the capacity to play this triple role, the expression of galectins in HNSCC tissue was analyzed in relation to HPV and *TP53* mutational status. Subsequently, the prognostic potential of each of the galectins in HNSCC was compared to identify the most promising panel of galectins as prognostic biomarkers for HNSCC. Finally, the ability of galectins to predict the response of HNSCC patients to standard treatments for this type of cancer was evaluated. This multi-dimensional approach provides insights into the intricate role of galectins in HNSCC, contributing to improving the prognosis and boosting the development of personalized treatment strategies.

## 2. Methods

### 2.1. Literature Survey of the Galectin Role in HNSCC Prognosis

A literature search was performed using a text-mining approach to ensure a comprehensive analysis of the existing literature. This search was performed in Web of Science, Scopus, and PubMed Databases between February 2022 and October 2023 using the following keywords [(galectin OR gal OR lgals) AND (squamous AND cell AND carcinoma)]. All original papers published in the last ten years were considered. From the literature search, 52 articles were selected. After analyzing the selected articles, we verified that 16 articles compared survival outcomes in HNSCC patients. These studies were integrated in our study. This literature search aimed to assess the state of the art regarding each of the galectins in the prognosis of head and neck cancer.

### 2.2. Galectins Expression and Survival in HNSCC

UALCAN Database (ualcan.path.uab.edu/home, accessed on 30 November 2023), Cancer Proteome and Phosphoproteome Atlas (CPPA) (http://cppa.site/cppa/, accessed on 30 November 2023), and ToPP Platform (http://www.biostatistics.online/topp/index.php, accessed on 30 November 2023) were assessed to evaluate the expression of galectins in HNSCC and healthy individuals, as well their impact on survival [43,44]. The web resources allow a comprehensive analysis of cancer Omics data (TCGA, MET500, and CPTAC). For these analyses, the “HNSCC-Head and Neck Squamous Cell Carcinoma” Dataset was used. Student’s t-test was used to calculate the *p*-value with a cutoff of 0.05.

### 2.3. Single-Cell RNA seq (scRNA-seq) Analysis of Galectins in HNSCC for Immune Cell Infiltration Analysis

Integration of the expression of each of the galectins in the HNSCC tumor microenvironment was done using the Tumor Immune Single Cell Hub 2 (TISCH2), ToPP, and canSARblack web tools [44,45,46]. TISCH2 (http://tisch.comp-genomics.org/, accessed on 30 November 2023) is a source of single-cell RNA-seq data that combines the information extracted from 190 tumor datasets across 50 cancer types. In this work, TISCH2 was used to evaluate the expression of galectins in various HNSCC cell types. The datasets that were evaluated are described in Appendix A. Using canSARblack (https://cansar.ai, accessed on 30 November 2023), the expression profile of each of the galectins in HNSCC was studied according to each one of the six immune phenotypes: C1 (wound healing), C2 (IFN-γ dominant), C3 (inflammatory), C4 (lymphocyte depleted), C5 (immunologically quiet), and C6 (TGF-β dominant). To evaluate the impact of galectin expression of C1 and C2 immune phenotypes of HNSCC, the ToPP platform was used. Survival analyses were conducted only on these two immune subtypes because they were the phenotypes where galectins were shown to be most overexpressed. For a deeper understanding of how single nucleotide variation (SNVs) and copy number variation (CNVs) influence immune infiltration, Gene Set Cancer Analysis (GSCA; http://bioinfo.life.hust.edu.cn/GSCA/#/, accessed on 30 November 2023) was used. To assess an association between immune cell infiltrates and galectins, the “Immune” module was used through ImmuCellAI to realize an association of genes with about 24 types of immune cells using the Wilcoxon test (comparison of two groups) or the One-Way ANOVA test (more than two groups). The *p*-value was adjusted by FDR.

### 2.4. Mutations and Post-Translation Modifications (PTM) for Each Galectin in HNSCC

TCGA-HNSC Dataset was used to identify CNV and SNV, and the methylation profile of the genes encoding galectins was assessed using GSCA and cBioPortal (https://www.cbioportal.org, accessed on 30 November 2023). cBioPortal is a web server used as a cancer genomics database based on TCGA. In this study, cBioPortal allowed a better understanding of the PTM associated with each galectin in HNSCC. The results obtained with cBioPortal were complemented with GSCA. In GSCA, the SNV data of 10,234 samples from 33 cancer types and CNV data of 11,495 samples were downloaded from TCGA Database. Methylation data of 14 cancer types were downloaded from Illumina Human Methylation 450k level 3 (TCGA Database) using the methylation module from GSCA. The Kaplan-Meyer survival analysis with log-rank test in GSCA was used to assess the correlation between the galectins that present these types of gene alterations with survival. The data were analyzed using R package survival, Cox Proportional-Hazards, and Logrank test.

### 2.5. Galectin Interaction Networks

GeneMANIA (https://genemania.org, accessed on 30 November 2023) is a web tool used for predicting gene function. This tool finds the most related genes to the input set of genes using a large set of functional association data. The gene-gene interaction network was constructed using this tool [47]. STRING database (https://string-db.org, accessed on 30 November 2023) was used for the analysis of the protein-protein interaction network. STRING provides associations between proteins using a score based on several sources such as scientific literature, experiments, computational interaction predictions, and systematic transfers of evidence among organisms [48]. REACTOME database (https://reactome.org, accessed on 30 November 2023) web tool allows the establishment of functional relationships between gene expression profiles, and it was used as a complementary tool for the functional enrichment of the main galectin interactors [49].

### 2.6. Galectins as Drug Targets in HNSCC

The potential of each of the galectins as the biomarker of response to the HNSCC treatment was studied using the GSCALite web server. GSCALite (http://bioinfo.life.hust.edu.cn/web/GSCALite/, accessed on 30 November 2023) is a web server that allows an integrated and broad approach to a set of genes [50]. In this case, GSCALite was chosen rather than GSCA; since GSCALite provides the values of a much larger number of drugs whereas GSCA only shows the results for the 30 drugs with the highest correlation with the genes of interest. One of the modules allows the assessment of drug sensitivity for genes using the Genomics of Drug Sensitivity in Cancer (GDSC) and Cancer Therapeutics Response Portal (CTRP) Databases. From GDSC, the inhibitory concentration 50% (IC50) of 265 small molecules in 860 cell lines and their corresponding mRNA gene expression were integrated, whereas the IC50 information of 481 small molecules in 1001 cell lines and their corresponding mRNA gene expression was extracted from CRTP. To establish the correlation between IC50 and corresponding mRNA gene expression, Pearson’s correlation was used and the *p*-value was adjusted by FDR.

## 3. Results

### 3.1. Identification of Galectin Isoforms with Key Role in HNSCC Pathogenesis

To assess the state of the art regarding the potential of galectins as prognostic biomarkers for head and neck cancer, a literature search was performed and the several survival-related parameters described in the articles were summarized in Table 1. It was possible to observe that in the existing literature, the most studied galectin was *Gal-3*. From the literature, we found a lot of heterogeneity in terms of the survival parameters analyzed. The most studied subtype of HNSCC was OSCC. By analyzing the Kaplan-Meyer curves associated with each of the galectins shown in Appendix A, we could observe that *Gal-1* and *Gal-2* have the greatest impact on the OS of patients with HNSCC. According to the TCGA analysis in UALCAN, *Gal-1* was shown to be overexpressed in HNSCC. *Gal-2* did not show significantly different expression compared to healthy individuals, as shown in Appendix A.

The proteins encoded by the galectin genes were characterized in terms of their impact on survival in this patient population using CPPA as shown in Figure 1 [5,6,7,8,9,10,11,12,13,14,15,16,17,18,19,20,21,22,23,24,25,26,27,28,29,30,31,32,33,34,35,36,37,38,39,40,41,42]. In CPPA, it could be seen that Gal-4 showed the highest HR relative to other galectins (HR = 14.848, *p* = 1.15 × 10^−5^). Gal-3BP was the second galectin with the highest impact on OS in HNSCC (HR = 9.605, *p* = 3.024 × 10^−4^), and its profile overlaps with the survival analysis performed in ToPP for the gene encoding Gal-4. CPTAC analysis was conducted to assess the galectin protein expression in HNSCC. We found that all proteins of interest were identified in HNSCC except for Gal-2, Gal-3BP, Gal-7C, Gal-9C, and Gal-S12, as shown in Figure 2. Using CPTAC, we analyzed the protein expression levels in HNSCC tissue compared to normal tissue, and we were able to see that for the galectins with higher impact on HNSCC survival, the following galectins Gal-1, Gal-3BP, and Gal-4 were overexpressed in HNSCC. Gal-3 was shown to be the most upregulated galectin in HNSCC (Figure 2). 

### 3.2. Identification of Galectins with Impact in the TME Remodeling

Galectins are involved in several processes related to the regulation of cell-cell interactions in the tumor microenvironment. Cancer cells can interact with various cell types that are part of the tumor microenvironment via galectins to create a more favorable environment for tumor invasion. Abnormal expression of galectins is associated with tumor development. To better understand which ligands and cell types bind to the galectins to support this favorable response to tumor progression, Figure 3 summarizes the key players in this response. When studying the immune subtypes of HNSCC, we found that Gal-1, Gal-3, Gal-3BP, Gal-4, and Gal-9 were mostly expressed in the IFN-γ Dominant (C2) immune subtype and to a lesser extent in the Wound Healing (C1) immune subtype, as shown in Table 2. Interestingly, the first four galectins were also the ones with a major impact on survival. Subsequently, the expression of galectins in the tumor microenvironment (TME) was characterized to understand how the TME can determine the expression of each one of the galectins. The results extracted from the TISCH2 web tool for each galectin and each GEO dataset are represented in Appendix A and the main results are illustrated in Appendix A. Among all galectins evaluated, Gal-1 had the highest expression levels in the evaluated cell types, including Tprolif, Treg, TCD8, monocytes/macrophages, dendritic cells, natural killer (NK) cells, myofibroblasts, and fibroblasts. Gal-3 and Gal-3BP were upregulated mostly in TCD8 cells, malignant cells, myofibroblasts, and fibroblasts. Gal-9 appeared selectively upregulated in dendritic cells and monocytes/macrophages, over other galectins that were unchanged. When assessing the impact of galectin expression in HNSCC according to immune subtype, it was observed that galectin expression in HNSCC cells with an IFN-γ Dominant (C2) profile had the highest impact on survival. Among the galectins expressed in HNSCC cells with this immune phenotype, *Gal-1*, *Gal-3BP*, and *Gal-8* were the galectins showing a higher HR value in survival analyses, as shown in Appendix A.

The most frequent PTM among all galectins was phosphorylation and the galectin that was targeted the most by PTM was *Gal-1*, as shown in Figure 4A. When analyzing the mutational profile of each of the galectins in HNSCC (Appendix A and Figure 4B), it was possible to observe that the most mutated galectins in HNSCC were *Gal-3BP*, *Gal-12*, *Gal-1*, and *Gal-4*. The results for the SNV impact on HNSCC survival were not statistically significant (Figure 4C). Of these mutations, most are missense. *Gal-8* transcription levels were the most influenced by CNV (Figure 4D). Regarding the methylation profile analysis of galectin genes in HNSCC, *Gal-12* was the most differentially methylated galectin in HNSCC compared to healthy individuals (Figure 4E). The galectin whose expression was most affected by the methylation process is *Gal-7* (Figure 4F). In Figure 4I, we can observe that in the HNSCC wild type group, there is a predominance of monocytes and neutrophils, whereas in the groups expressing galectins that present CNV alterations, there is a predominance of CD8 T cells, NK cells, follicular helper T CD8 cells, and cytotoxic T cells. When the groups in which there is gene amplification are compared to the group in which there is gene deletion, the main difference is the inversion of the proportion of follicular T CD4 cells and NK cells. In the group with galectin gene amplification, the proportion of follicular helper T CD4 cells is higher than NK cells, whereas in the group with galectin gene deletion, there are more NK cells. In Figure 4J, it was possible to observe that in the group of galectins presenting SNV, the immune cell abundance was lower than that in the wild type. The galectins with CNV changes that showed a higher HR in terms of PFS were *Gal-4*, *Gal-7*, and *Gal-7B* (Figure 4G). When the differences in survival were compared according to the methylation profile of each of the galectins, it was possible to observe that the methylation of *Gal-7* was shown to correlate with the highest HR of the galectins studied in terms of DFI (Figure 4H). Furthermore, we used GeneMANIA to obtain potential interaction genes of the galectin family. Figure 4K shows that the galectin genes interact with several genes of the pentraxin family that encode acute-phase inflammatory proteins. The protein-protein networks are shown in Figure 4L, M. It could be seen that the proteins that bind most strongly to the galectins are CD44 (CD44 antigen), HRAS (GTPase HRas), EGFR (Epidermal growth factor receptor), HAVCR2 (Hepatitis A virus cellular receptor 2), and CALCOCO2 (Calcium-binding and coiled-coil domain-containing protein 2). By analyzing these 5 interactors using the REACTOME database, it was possible to observe that they essentially intervene in the EGFR and ERBB2 signaling pathways. *Gal-1* and *Gal-3BP* showed to be the most interesting galectins in HNSCC. These galectins are involved in several processes such as migration, invasion, and angiogenesis of HNSCC, as described in Figure 5.

### 3.3. Galectins and HNSCC Response to Therapeutics

According to the National Cancer Institute, there are several drugs approved for head and neck cancer, such as bleomycin sulfate, cetuximab, docetaxel, hydroxyurea, methotrexate sodium, pembrolizumab, and nivolumab. Carboplatin with docetaxel and cisplatin with docetaxel and 5-fluorouracil are approved as drug combinations. To evaluate the correlation between the galectins and drugs present in the GDSC and CTRP databases, GSCALite was used. A summary of galectin interaction results with FDA-approved drugs for HNSCC is shown in Table 3. Bleomycin showed a negative Pearson’s correlation with *Gal-1* and cetuximab showed a negative correlation with *Gal-1*, *Gal-3*, and *Gal-3BP* expression. 5-fluouracil, methotrexate, and paclitaxel were positively correlated with *Gal-3* and *Gal-3BP* expression. 5-fluouracil and paclitaxel were also positively correlated with *Gal-8* expression. 

## 4. Discussion

In the present work, the role of galectins was comprehensively studied to compare the contribution of each galectin in the processes of tumor progression and invasion, as well as the added value they may have in the early diagnosis of HNSCC and the evaluation of treatment response. To this end, a literature search was initially conducted to analyze the evidence that exists for these proteins in HNSCC. Most of the published articles focused on the impact of galectin expression on prognosis, with *Gal-3* being the one with the most literature evidence. Taking into account the existing information for *Gal-3* in HNSCC, a systematic review was done in PROSPERO with the following number CRD42023400863. Subsequently, a bioinformatics analysis was conducted, and the expression levels and impact on the prognosis of genes encoding galectins and proteins were assessed using platforms such as ToPP, UALCAN, and CPPA. Based on the results obtained, it was possible to observe that *Gal-1*, *Gal-3*, *Gal-3BP*, *and Gal-4* proved to be the most impactful galectins in the prognosis of HNSCC. The results obtained from the bioinformatics analysis provide supporting evidence for the existing literature, while also paving the way for future research. One of the least studied galectins, *Gal-3BP*, has been found to have a significant impact on the prognosis of HNSCC. The *LGALS3BP* gene exhibited an HR = 2.6 and the Gal-3BP protein an HR = 9.605 when evaluating the impact of their expression on the OS of HNSCC patients. High levels of *Gal-3* expression in HNSCC were associated with increased progression, invasiveness, and aggressiveness in part by the activation of pathways such as the Wnt/B-catenin, ERK1/2, and AKT pathways. *Gal-3* expression is dependent on the cell differentiation process in both normal and neoplastic cells [12,21,22,23,32,34,38,40,41,103,104]. Colocalization of Gal-3 with desmosomal proteins demonstrated that this protein may play an important role at the cell surface level in mediating intercellular contacts between tumor cells. Thus, it can be used to monitor the degree of cell differentiation in carcinomas that have their genesis in neoplastic transformation cells [11,13,30]. Studies have shown that serum levels of Gal-3 are increased in individuals with HNSCC compared to those of controls, and its expression in individuals with risk factors for this type of cancer has been associated with an approximately 3-fold increased risk of developing HNSCC [6,105]. Thus, it has the potential to be used in HNSCC screening using serum as a source for liquid biopsy. Regarding the treatment, Gal-3 inhibition has shown potential as a therapeutic option in these patients, especially in HPV-driven HNSCC which presents a much higher Gal-3 expression than in HPV-non-driven HNSCC patients, due to the inhibition of this protein particularly interesting in this cancer subset [37]. *Gal-3* seems to have an immunosuppressive effect by promoting M2 macrophage polarization, inducing lymphocyte apoptosis, and inhibiting T-cell activation that contributes to tumor progression. The goal of Gal-3 inhibitors is to restore the immune response that is suppressed by Gal-3 [37,106]. On ClinicalTrials.gov, there are 3 clinical trials with Gal-3 inhibitors for the treatment of HNSCC, namely, NCT00054977, NCT02575404, and NCT04987996. Clinical trial NCT00054977 is a phase I, open-label, and non-randomized trial that has been completed and aimed to study the safety of GM-CT-01 alone and in combination with 5-fluorouracil in HNSCC patients. The results have not been published. NCT04987996 is a phase II randomized trial that aims to evaluate the safety and efficacy of Gal-3 inhibitor, GR-MD-02, compared with pembrolizumab in the treatment of HNSCC. This study is currently on hold. NCT02575404 is an ongoing phase I, open-label, and non-randomized clinical trial that aims to evaluate the dose escalation of GR-MD-02 with pembrolizumab in patients with HNSCC. More studies are still needed, but the development of Gal-3 inhibitors that can be administered alone or in conjunction with standard HNSCC treatments to increase their effectiveness is shown to be a promising strategy. By studying the role of each of the galectins in the tumor microenvironment, *Gal-1*, *Gal-3*, and *Gal-3BP* have been shown to be highly overregulated in their expression levels in the major cells that constitute the tumor microenvironment. The development of drugs that inhibit the overexpression of these 3 galectins will allow modulation of the tumor microenvironment, increasing the sensitivity of patients to immunotherapy treatments.

Studies have shown that the high expression of Gal-3BP, compared to other galectins, is associated with a worse prognosis, including lower overall survival, disease-free survival, and relapse-free survival. This protein’s involvement in the PI3K/AKT pathway contributes to its impact on tumor progression [107]. Gal-1 is a prototypical galectin that modulates the process of differentiation, immune escape, and tumor progression. Gal-1 levels are modulated by blood oxygen levels. When in hypoxia, Gal-1 expression increases, and this increased expression is associated with the secretion of proteins that modulate the immune response, assuming an active role in the malignant progression and therapeutic response of HNSCC [9]. Considering the protective effect that Gal-1 exerts on cancer cells against the action of the immune system through its deleterious effect on activated T-cells and the activation of oncogenic H-Ras proteins, the ways to reduce its expression to increase the efficiency of T-cell mediated immunotherapy in patients with HNSCC have also been evaluated [9,26,31]. Therefore, high levels of Gal-1 are associated with a worse prognosis of HNSCC. Gal-1 is involved in tumor invasion and metastasis processes. This role is due in part to the increase in certain metalloproteinases in response to increased Gal-1 expression levels and the fact that it is involved in the reorganization of the cytoskeleton in oral cancer [14,16,29,35]. Increased Gal-7 levels have been shown to be associated with tumor progression, a higher recurrence rate, and a worse prognosis of HNSCC [10,18,42]. Gal-8 and Gal-9 were shown to have the potential to differentiate HNSCC from other potentially malignant oral lesions and healthy tissues [17,27,28,60]. The work done in this study allowed a more comprehensive and detailed characterization of the biomarkers, thereby identifying those with the greatest potential for translation into clinical practice.

## 5. Conclusions

Galectins play a very important role in tumor differentiation, progression, and invasion, being associated with a worse prognosis of HNSCC. *Gal-3* and *Gal-1* are the galectins with more literature evidence pointing to their potential as prognostic biomarkers. From the bioinformatics analysis, *Gal-1* and *Gal-3BP* were shown to be strongly modulated by HPV, *TP53* mutational status, and tumor microenvironment. In addition, they have been shown to be valuable biomarkers of prognosis and treatment response in patients with HSNCC. Future research should prioritize elucidating the intricate role of *Gal-3BP* in HNSSC, as well as investigating the link between the expression of each of the galectins and HPV and *TP53* mutational status. Such investigations hold significant promise for advancing our understanding of HNSCC pathogenesis, improving prognostic assessment, and paving the way for targeted therapeutic interventions in this complex malignancy.

## Figures and Tables

**Figure 1 biomedicines-12-00529-f001:**
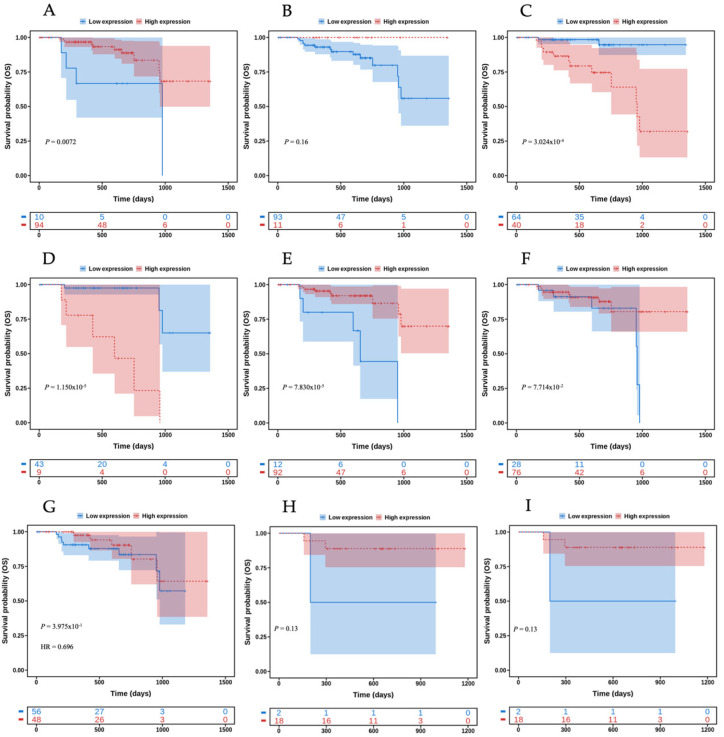
Survival analysis of galectin proteins on CPPA. Overall survival of LGALS1 (**A**), LGALS3 (**B**), LGALS3BP (**C**), LGALS4 (**D**), LGALS7 (**E**), LGALS8 (**F**), LGALS9 (**G**), LGALS9B (**H**) and LGALS9C (**I**) proteins in HNSCC. The hazard ratios for each galectin were 0.232 (**A**), 0.000 (**B**), 14.848 (**C**), 0.137 (**D**), 0.696 (**E**), respectively. The *X*-axis represents the survival time of the patients and the *Y*-axis represents the probability of survival.

**Figure 2 biomedicines-12-00529-f002:**
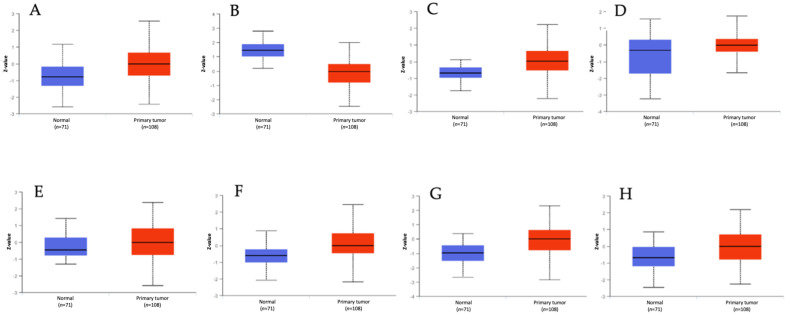
Galectin protein expression in HNSCC and healthy patients based on CPTAC. Box plot of the protein expression of LGALS1 (**A**), LGALS3 (**B**), LGALS3BP (**C**), LGALS4 (**D**), LGALS7 (**E**), LGALS8 (**F**), LGALS9 (**G**), and LGALS9B (**H**) in HNSCC (*n* = 71) and healthy tissue samples (*n* = 108). All results were statistically significant (*p* < 0.05).

**Figure 3 biomedicines-12-00529-f003:**
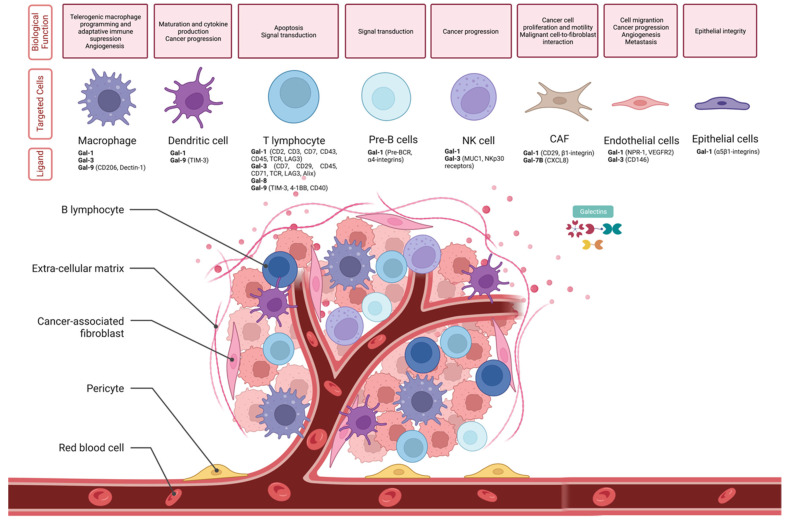
Relationship of galectins with tumor microenvironment. Biological function of galectins in each of the major cell types present in the tumor microenvironment [51,52,53,54,55,56,57,58,59,60,61,62,63,64,65,66,67,68,69,70,71,72,73,74,75,76,77,78,79,80,81,82,83,84,85,86,87,88,89,90,91,92,93,94,95,96,97,98,99,100,101,102]. Legend: CAF, cancer-associated fibroblasts; Gal-1, galectin-1; Gal-3, galectin-3; Gal-9, galectin-9; and NK cell, natural killer cell. This image was created in BioRender (Toronto, ON, Canada).

**Figure 4 biomedicines-12-00529-f004:**
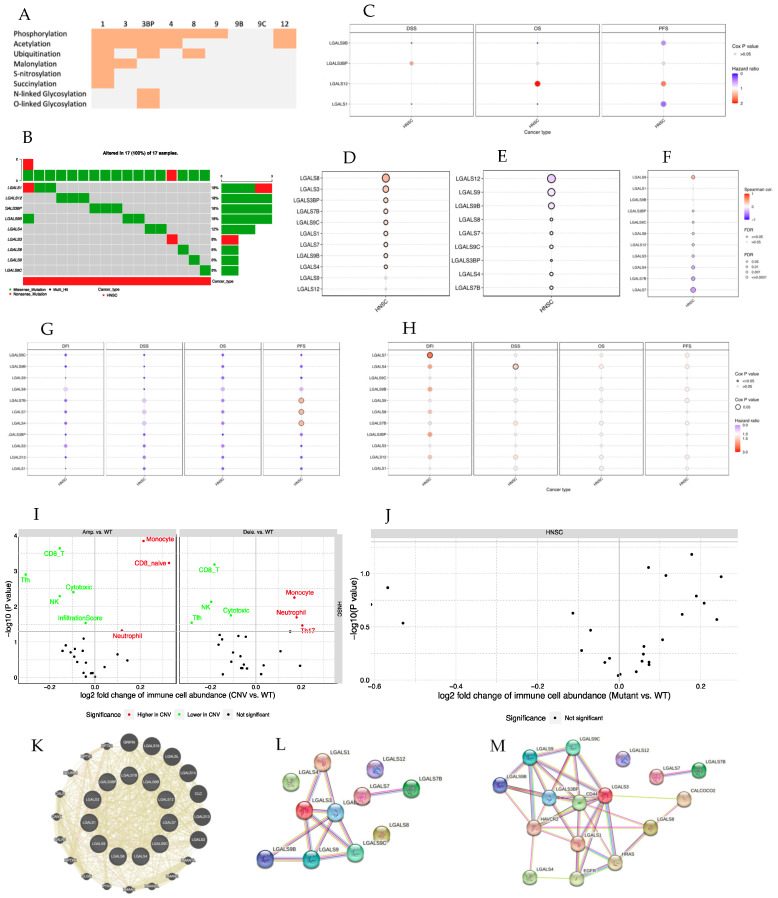
SNV, CNV, and PTM alterations of the galectins in HNSCC and their impact on survival. PTM of each galectin in HNSCC using cBioPortal (**A**). SNV frequency for each galectin in HNSCC using GSCALite (**B**). Survival analysis for galectins SNV in GSCA (**C**). Pearson correlation between expression of galectins and the corresponding CNV (**D**). Differential methylation between HNSCC and healthy tissue samples using GSCALite (**E**). Pearson correlation between expression of galectins and their methylation profile using GSCALite (**F**). Survival analysis for galectins CNV in GSCA (**G**). Survival analysis for high and low methylation groups in GSCA (**H**). Difference of immune infiltration between CNV of galectins and wild type HNSCC in GSCA (**I**). Difference of immune infiltration between SNV of galectins and wild type HNSCC in GSCA (**J**). Gene-gene interaction network of galectin of interest in GeneMANIA (**K**). Protein-protein interaction network analysis of galectins with STRING (**L**). Protein-protein interaction expanded network analysis of galectins with STRING (**M**). In GSCALite, only statistically significant results (*p* < 0.05) are represented. In the scatter plots represented in (**I**,**J**), the *p*-value statistically significant is represented with green color, and the FDR significant results are represented with red color. The set of genes used in GSCA and GSCALite analysis were LGALS1, LGALS12, LGALS3, LGALS3BP, LGALS4, LGALS7, LGALS7B, LGALS8, LGALS9, LGALS9B, LGALS9C, and LGALS12.

**Figure 5 biomedicines-12-00529-f005:**
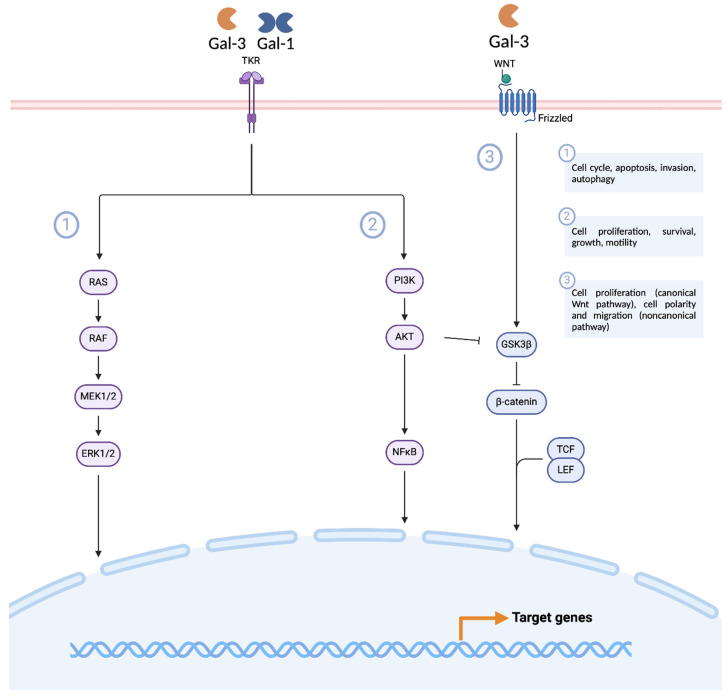
Overview of the main pathways modulated by the galectins, Gal-1 and Gal-3, found altered in HNSCC. Interaction with the PI3K/Akt pathway is responsible for the migration and proliferation of tumor cells through the activation of ß-catenin. When Gal-1 and Gal-3 interact with TKR, they stimulate the H-Ras pathway and promote cell proliferation. Legend: Akt, protein kinase B; ERK, extracellular signal-regulated protein kinase; GSK3ß, glycogen synthase kinase 3 beta; ILK, integrin-linked kinase; LEF, lymphoid enhancer-binding factor; MEK, mitogen-activated protein kinase; NF-KB, nuclear factor kappa-light-chain-enhancer of activated B cells; PI3K, phosphoinositide 3-kinase; RAF, rapidly accelerated fibrosarcoma; RAS, rat sarcoma; TCF, T cell factor; and WNT, Wingless-related integration site. This image was created in BioRender (Toronto, ON, Canada).

**Table 1 biomedicines-12-00529-t001:** Literature overview of the galectin impact on several parameters related to HNSCC prognosis. Influence of each galectin in tumor size (T), lymph node invasion (N), metastasis (M), invasion pattern (IP), relapse-free survival (RFS), disease-free survival (DFS), recurrence rate (RR), and histological grade malignancy (HGM) in head and neck squamous cell carcinoma (HNSCC). The analyzed HNSCC types are GSCC, gingival squamous cell carcinoma; LSCC, laryngeal squamous cell carcinoma; OSCC, oral squamous cell carcinoma; PSCC, palate squamous cell carcinoma; SS, sample size; TSCC, tongue squamous cell carcinoma. The studies with a statistically significant correlation (*p* < 0.05) with each one of the evaluated parameters are presented in green color and those without a statistically significant impact on survival-related parameters are presented in orange color.

Galectin	Tumor Site	SS	T	N	M	IP	OS	RFS	DFS	RR	HGM
Galectin 1	GSCC	80									
	OSCC	64									
	LSCC	62									
	TSCC	65									
Galectin 3	OSCC	60									
	OSCC	60									
	OSCC	98									
	OSCC	32									
	OSCC/LSCC	53									
	LSCC	73									
	PSCC	45									
	TSCC	65									
Galectin 3BP	OSCC	92									
Galectin 4	TSCC	65									
Galectin 7	OSCC	32									
	HSCC	81									
	TSCC	65									
Galectin 8	HNSCC	93									
	LSCC	77									
Galectin 9	OSCC	32									

**Table 2 biomedicines-12-00529-t002:** Expression of galectins in each immune subtype in HNSCC Dataset in canSAR.ai.

Cancer Immune Subtype	Gal-1	Gal-3	Gal-3BP	Gal-4	Gal-7, 7B	Gal-8	Gal-9	Gal-9B	Gal-9C	Gal-12
Wound Healing (C1)	29/128	5/128	0/128	8/128	0/128	1/128	1/128	0/128	0/128	0/128
IFN-γ Dominant (C2)	93/379	13/379	10/379	14/379	0/379	1/379	25/379	0/379	0/379	0/379
Inflammatory (C3)	0/2	0/2	0/2	0/2	0/2	0/2	0/2	0/2	0/2	0/2
Lymphocyte Depleted (C4)	0/2	0/2	0/2	0/2	0/2	0/2	0/2	0/2	0/2	0/2
Immunologically Quiet (C5)	0/0	0/0	0/0	0/0	0/0	0/0	0/0	0/0	0/0	0/0
TGF-β Dominant (C6)	1/3	0/3	0/3	0/3	0/3	0/3	0/3	0/3	0/3	0/3

Legend: 1, galectin-1; 3, galectin-3; 3BP, galectin-3BP; 4, galectin-4; 7, galectin-7; 7B, galectin-7B, 8, galectin-8; 9, galectin-9; 9B, galectin-9B; 9C, galectin-9C; and 12, galectin-12. The values in the table refer to the number of samples where the expression of each of the galectins is seen relative to the total number of samples of each immune phenotype in HNSCC.

**Table 3 biomedicines-12-00529-t003:** Galectin-drug interaction using GSDC and CTRP Databases in GSCALite. Legend: CTRP, Genomics of Drug Sensitivity in Cancer; FDR, False Discovery Rate; and CTRP, Cancer Therapeutics Response Portal. Blue represents a negative Spearman correlation between the gene and the chosen drug. Red represents a positive Spearman correlation. The results shown in this figure were statistically significant (*p* < 0.05).

Databases	Drugs	LGALS1	LGALS3		LGALS3BP	LGALS4	LGALS8	LGALS9	LGALS12
GSDC	Bleomycin	log_10_(FDR) = 20							
	5-fluorouracil		log_10_(FDR) = 10		log_10_(FDR) = 10				
	Methotrexate		log_10_(FDR) = 40		log_10_(FDR) = 40		log_10_(FDR) = 10		
	Docetaxel	log_10_(FDR) = 20	log_10_(FDR) = 10		log_10_(FDR) = 10				
	Cetuximab		log_10_(FDR) = 10						
CTRP	Docetaxel		log_10_(FDR) = 20		log_10_(FDR) = 20	log_10_(FDR) = 10			
	Paclitaxel		log_10_(FDR) = 30		log_10_(FDR) = 30		log_10_(FDR) = 10		

## Data Availability

The datasets analyzed during the current study are available in the Proteomics Identifications Database (https://www.ebi.ac.uk/pride/) and Gene Expression Omnibus (https://www.ncbi.nlm.nih.gov/gds/) repositories.

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
