# Peer review of "Revisiting Multi-Omics Data to Unravel Galectins as Prognostic Factors in Head and Neck Squamous Cell Carcinoma"

_biomedicines, 2024, doi:10.3390/biomedicines12030529_

Round 1

Reviewer 1 Report

Comments and Suggestions for Authors

Brief summary

The paper focus the attention on a very interesting topic, as galectin family expression is widely studied in normal and pathological condition, mostly in non neoplastic diseases, as in cardiac and liver pathology. However, these proteins play a role in tumour microenvironment, which is now considered an hot topic in cancer pathology.

It is an original production, with clear structured and solid analysis of the literature.

Conclusions are consistent with the thesis and argument presented.

No ethical problems are found in this study

I would like to make some suggestions and I have few questions

General concept comments

have you analyse the whole literature present on galectin expression ?
I suggest to add these information in the method part, as it is not clear of you have a cuf off for old paper or you analyse all of them

Author Response

The paper focus the attention on a very interesting topic, as galectin family expression is widely studied in normal and pathological condition, mostly in non neoplastic diseases, as in cardiac and liver pathology. However, these proteins play a role in tumour microenvironment, which is now considered an hot topic in cancer pathology.

It is an original production, with clear structured and solid analysis of the literature.

Conclusions are consistent with the thesis and argument presented.

No ethical problems are found in this study

I would like to make some suggestions and I have few questions

General concept comments

have you analyse the whole literature present on galectin expression? I suggest to add these information in the method part, as it is not clear of you have a cuf off for old paper or you analyse all of them.

R: Following Reviewer’s concerns, we improved the description of the literature survey performed in subsection 4.1. from Methods section. In fact, we used a rigorous text-mining approach to ensure a comprehensive analysis of the existing literature. This approach allowed us to search multiple databases, including PubMed and Scopus, using specific search terms related to galectins and HNSCC prognosis. We carefully selected the last 20 years to include current studies to ensure a comprehensive understanding of the topic. We focused on studies that explicitly investigated the impact of galectins on the prognosis of HNSCC patients so that we could provide a detailed analysis in our results.

Reviewer 2 Report

Comments and Suggestions for Authors

It is necessary to comprehend the role of galectins in HNSCC, contributing to improve the prognosis and boost the development of personalized treatment strategies by multi-dimensional approach. Some concerns should be addressed before publication.

1.     Title: Be sure the title includes any specific terms as directed in the reporting guidelines for your type of article (for example, "case report" should be in the title of a CARE-compliant article). The following guidelines specify terms that should be in the title: CARE, CHEERS, CONSORT, PRISMA.

2.     This is a database secondary analysis. Why the authors didn’t conduct a meta-analysis for effect of galectins in HNSCC after all reviewing relevant literatures.

3.     How to access this database? The authors should address this concern.

4.     Could the authors add some mechanism plot of galectins in HNSCC to meet the merit of Biomedicnes.

5.     Some references should be updated.

Author Response

Comments and Suggestions for Authors

It is necessary to comprehend the role of galectins in HNSCC, contributing to improve the prognosis and boost the development of personalized treatment strategies by multi-dimensional approach. Some concerns should be addressed before publication.

  1. Title: Be sure the title includes any specific terms as directed in the reporting guidelines for your type of article (for example, "case report" should be in the title of a CARE-compliant article). The following guidelines specify terms that should be in the title: CARE, CHEERS, CONSORT, PRISMA.

R: We understand reviewer’s point of view, but our paper does not fall under the specific categories of case reports, economic evaluations, randomized trials, or systematic reviews. Therefore, the aforementioned guidelines (CARE, CHEERS, CONSORT, PRISMA) are not directly applicable to our manuscript type. Nevertheless, we changed the title to “Revisiting Multi-Omics Data to Unravel Galectins as Prognostic Factors in Head and Neck Squamous Cell Carcinoma”, to better reflect the fact that we revisited Omics data focusing on the putative role of galectins.

  1. This is a database secondary analysis. Why the authors didn’t conduct a meta-analysis for effect of galectins in HNSCC after all reviewing relevant literatures.

R: Indeed, we conducted a secondary analysis of Omics data retrieved from the selected papers using bioinformatic tools. We would like to clarify that our decision to conduct a secondary database analysis rather than a direct meta-analysis was a strategic decision consistent with our research objectives. Our aim was to conduct a multi-faceted investigation of the expression and prognostic value of galectins in HNSCC. This included not only examining survival rates, but also exploring their relationship to key factors such as HPV and TP53, as well as the influence of the tumor microenvironment on galectin expression and its subsequent impact on prognosis. This comprehensive approach has allowed us to characterize the role of galectins in HNSCC in more detail and comprehensively. We believe that this in-depth analysis provides different insights than the ones given by a meta-analysis. We appreciate your suggestion and would like to assure you that a meta-analysis is indeed part of our future research plan. It is currently registered in PROSPERO and will complement our previous findings with a broader, quantitative synthesis of the available data. We hope that we have clarified the rationale behind our methodological choices. We thank you for your insightful feedback and are open to further discussion.

  1. How to access this database?

R: Thank you for your comment and the opportunity to clarify our methodological strategy. In our research, we accessed each of the following databases individually to collect specific information for each gene of interest:

  1. UALCAN (http://ualcan.path.uab.edu/home)):  

   - Overview: An interactive web portal for in-depth analysis of TCGA gene expression data.  

   - Output: Provided expression levels of galectin genes in HNSCC and normal tissues, enabling comparison and analysis of differential expression.

  1. Cancer Proteome Atlas (CPPA) (http://cppa.site/cppa/)):  

   - Overview: A resource for exploring cancer proteomic data.  

   - Output: Proteomic data related to galectins in HNSCC, aiding in understanding protein level changes.

3.BioStatistics Online ([ (http://www.biostatistics.online/topp/index.php)):  

   - Overview: Offers statistical tools and resources for biomedical research.  

   - Output: Statistical analysis and trends of galectin gene expression, providing a quantitative basis for their significance in HNSCC.

  1. TISCH http://tisch.comp-genomics.org/)):  

   - Overview: A comprehensive resource for tumor-immune system interactions, with single-cell RNA-sequencing data.  

   - Output: Single-cell RNA-sequencing data of galectin genes in HNSCC, offering insights into their role at a cellular level.

  1. Gene Set Cancer Analysis (GSCA) http://bioinfo.life.hust.edu.cn/GSCA/#/)):  

   - Overview: Integrates gene set analysis in various cancers.  

   - Output: Integrated analysis of galectin gene sets in cancer, including HNSCC, for a broader understanding of their roles in cancer biology.

  1. cBioPortal (https://www.cbioportal.org)):  

   - Overview: A comprehensive web resource for exploring, visualizing, and analyzing cancer genomics data.  

   - Output: Genetic alterations of galectins in HNSCC datasets, useful for understanding the genetic landscape.

  1. GeneMANIA (https://genemania.org)):  

   - Overview: Predicts gene function and generates hypotheses about gene-gene interactions.  

   - Output: Network analysis of galectin genes and their interactions, providing insights into their functional relationships.

  1. STRING (https://string-db.org)):  

   - Overview: A database of known and predicted protein-protein interactions.  

   - Output: Protein-protein interaction networks for galectins, aiding in understanding their interactions and functional associations.

  1. Reactome (https://reactome.org)):  

   - Overview: A curated database of pathways and reactions in human biology.  

   - Output: Pathway analysis involving galectin genes, providing insights into their biological roles and pathways.

10.GSCALite (http://bioinfo.life.hust.edu.cn/web/GSCALite/)):  

    - Overview: A web tool for gene set cancer analysis, including gene expression, regulation, and survival.  

    - Output: Analysis of galectin gene expression and its effect on survival in cancer, including HNSCC, contributing to prognostic understanding.

For each database, we extracted relevant data related to galectin genes associated with HNSCC. This allowed us to collect a comprehensive dataset, which we then carefully compiled to analyze the prognostic significance of galectins in HNSCC. The detailed methodology, including the specific data we extracted from each database, is described in the Methods section. We believe that this methodological approach increases the robustness and depth of our results. We appreciate your inquiry as it highlights an important aspect of our research methodology.

  1. Could the authors add some mechanism plot of galectins in HNSCC to meet the merit of Biomedicines.

R: Following Reviewer’s suggestion, we have developed a comprehensive mechanism plot to supplement our manuscript (Figure 6). This diagram meticulously illustrates the pathways involving galectins in HNSCC and includes their influence on cell proliferation, apoptosis, angiogenesis, immune response and metastasis. We have used various visual elements and clear annotations to ensure that the presentation is not only informative but also accessible to readers with different backgrounds. Our goal with this supplementary figure is to provide a concise yet comprehensive overview of galectin interactions in HNSCC.

  1. Some references should be updated.

R: Following Reviewer’s concern the references were updated.